# Role of a Polyphenol-Enriched Blueberry Preparation on Inhibition of Melanoma Cancer Stem Cells and Modulation of MicroRNAs

**DOI:** 10.3390/biomedicines12010193

**Published:** 2024-01-16

**Authors:** Nawal Alsadi, Nour Yahfoufi, Carolyn Nessim, Chantal Matar

**Affiliations:** 1Department of Cellular and Molecular Medicine, Faculty of Medicine, University of Ottawa, 451 Smyth Road, Ottawa, ON K1H 8M5, Canada; nalsa068@uottawa.ca (N.A.); nyahf074@uottawa.ca (N.Y.); 2Department of Surgery, University of Ottawa, 501 Smyth Road, Ottawa, ON K1H 8L6, Canada; cnessim@toh.ca; 3School of Nutrition Sciences, Faculty of Health Sciences, University of Ottawa, 451 Smyth Road, Ottawa, ON K1H 8M5, Canada

**Keywords:** polyphenol, miRNA, EMT, NF-κB, metastasis, melanoma cancer stem cells

## Abstract

Melanoma is a type of skin cancer known for its high mortality rate. Cancer stem cells (CSCs) are a subpopulation of cancer cells that significantly contribute to tumour recurrence and differentiation. Epigenetic-specific changes involving miRNAs maintain CSCs. Plant polyphenols have been reported to be involved in cancer chemoprevention and chemotherapy, with miRNAs being the novel effectors in their biological activities. A polyphenol-enriched blueberry preparation (PEBP) derived from fermented blueberries has demonstrated promising chemopreventative properties on breast cancer stem cells by influencing inflammatory pathways and miRNAs. In our current investigation, we seek to unveil the impact of PEBP on inhibiting melanoma development and to elucidate the underlying mechanisms. Our study employs various human cell lines, including an ex vivo cell line derived from a patient’s metastatic tumour. We found that it elevates miR-200c, increasing E-cadherin expression and inhibiting miR-210-3p through NF-κB signalling, impacting Epithelial-to-Mesenchymal Transition (EMT), a critical process in cancer progression. PEBP increases the SOCS1 expression, potentially contributing to miR-210-3p inhibition. Experiments involving miRNA manipulation confirm their functional roles. The study suggests that PEBP’s anti-inflammatory effects involve regulating miR-200c and miR-210 expression and their targets in EMT-related pathways. The overall aim is to provide evidence-based supportive care and preclinical evaluation of PEBP, offering a promising strategy for skin cancer chemoprevention.

## 1. Introduction

Malignant melanoma is one of the most common, aggressive, and drug-resistant skin cancer diseases worldwide [1,2]. Currently, a range of treatment options exists, demonstrating efficacy in approximately 50% of patients. However, a significant number remain unresponsive to these interventions, ultimately succumbing to the disease. Therefore, the imperative arises to explore alternative strategies for the treatment of this condition. Malignant melanoma is known to have a poor prognosis, due to the potential for vascular invasion, metastasis, and recurrence [2,3,4]. A subset of neoplastic cells shares some characteristics with normal stem cells, such as self-renewing and differentiation, and these are termed cancer stem cells (CSCs) [5]. Conventional anti-cancer drugs and radiotherapy have the ability to eradicate the bulk of tumour mass only [5,6]. However, there is little or no effect on melanoma CSCs, which will lead to tumour recurrence and progression [3,7]. Hence, more research is needed to fully understand the mechanisms for preventing metastasis in melanoma CSCs and to develop effective targeted therapies.

Recent studies on the acquisition of mesenchymal traits by epithelial cancer cells during tumour invasion and metastasis have provided a better understanding of the metastatic progression and its mechanisms [8]. The epithelial–mesenchymal transition (EMT) process changes the phenotypes of the cells by altering adhesive properties with adjacent cells [8,9]. These cells gain motility and migrate throughout territories distant from the primary tumour. These mechanisms involve several cellular signalling pathways, such as erythroblastic leukaemia viral oncogene homolog (ErbB), Wingless/Integrated (Wnt), Nuclear factor kappa-light-chain-enhancer of activated B cells (NF-κB), and transforming growth factor-beta (TGF-β) [8,10,11,12]. Among these, the most critical pathway in inducing the EMT process is NF-κB, which activates the expression of transcription factors, Snail1, Twist1, and ZEB1/2 [10,13]. Although melanocytes are derived from neural crest cells and are not epithelial in origin, EMT is a well-documented phenomenon contributing to the metastatic potential of malignant melanocytes [8,9].

A growing body of evidence has shown that epigenetic modification plays a critical role in contributing to CSC characteristics, including self-renewal ability, and differentiation mediated by miRNAs [14]. Several studies have been published on the effects of miRNAs on metastasis and invasion [9,15]. Metastasis-specific miRNA signatures directly compared primary tumours and metastases in lymph nodes. This showed a differential expression of key miRNAs responsible for clinical and pathological aspects of many tumours [6,16,17]. miRNAs are involved in cancer initiation, progression, and the maintenance of stemness status [17,18]. More precisely, numerous scientific reports have identified the impact of polyphenols on regulating several human miRNAs that are dysregulated in cancer [17,18,19]. Studies have shown that polyphenolic compounds could revert the malignant transformation of cancer by restoring tumour suppressor miRNAs [20,21]. Previous studies by Mallet et al. showed that PEBP increased the expression of miR-145 and inhibited the level of miR-210 expression in 4T1 and MDA-MB-231 breast cancer cell lines [22]. Moreover, we also found an increase in the expression level of miR-200b in different skin cells exposed to PEBP [20], thus providing insights into the role of miRNAs in modulating cancer stem cells [18,19]. However, the protective role of polyphenolic compounds in modulating miRNAs in melanoma cancer stem cells is not yet fully understood.

Plants are a rich source of dietary phytochemicals endowed with chemoprevention potential against metastasis, migration, and cell invasion in melanoma and other type of cancer [23,24,25,26,27]. Polyphenols from fruits, vegetables, grape seeds, and green tea have been shown to protect the skin from the adverse effects of solar UV radiation [20,25]. Phytochemicals with anti-inflammatory, immunomodulatory, and antioxidant properties have the highest potential for exhibiting chemo-preventive behaviour in skin cancers [23,25]. Small fruits such as blueberries are known for their high concentration of phenolic compounds, including anthocyanins. The fermentation of blueberry preparation by probiotic bacterium (*SV-53*) yielded a preparation designed as the Polyphenol-Enriched Blueberry Preparation (PEBP), and increased the antioxidant, anti-inflammatory and anti-diabetic properties [20,24,25]. PEBP has been shown to decrease the formation of mammospheres in different cell lines and significantly reduce mammary carcinoma growth in mice. Furthermore, it reduces lung metastasis and controls the formation and proliferation of CSCs, while protecting neurons from oxidative stress caused by hydrogen peroxide [20,24].

The analysis and fractionation of extract from fermented blueberry or PEBP using ultra-performance liquid chromatography-quadrupole time-of-flight mass spectrometry (UPLC-MS-QTOF) have revealed that specific fractions are enriched with bioactive compounds such as gallic acid (GA), Protocatechuic acid (PCA), and catechin [24]. These compounds have been found to be effective in inhibiting metastasis and limiting the antioxidant capacity of cancer cells [24,28]. A balanced mixture of polyphenols, including GA, PCA, and catechins, was used to mimic the fermentation process, resulting in an Oligomeric Mixture of Polyphenols (OMP) [24].

The hypothesis stems from the observation that small oligomers of polyphenols found in the fermented mixture can modulate skin cancer stem cells through the regulation of microRNAs-related to inflammatory pathways involved in skin cancers. Hence, the aim of this study was to demonstrate the effect of polyphenolic compounds on skin cancer stem cells, microRNAs, and inflammatory pathways.

## 2. Materials and Methods

### 2.1. Preparation of Blueberry Mixture

Fully matured wild blueberries (*Vaccinium angustifolium* Ait.) were purchased from Cherryfield Foods Inc. as fresh and untreated fruits (Cherryfield, ME, USA). A total of 100 g of blueberries were blended using a Braun Type 4259 food processor then centrifuged at 500× *g* for 10 min in an IEC Centra MP4R centrifuge (International Equipment Company, Needham Heights, MA, USA). Finally, the juice was sterilized by filtration through a 0.22 µm Express Millipore filter apparatus (Millipore, Etobicoke, ON, Canada) to remove fruit skin and non-homogenized particles.

The bacterium *Rouxiella badensis* subsp *acadiensis SV-53*, also known as *Serratia vaccinii*, was cultured as previously described [29]. To begin the fermentation process, a saturated culture of the bacterium was added to the juice, constituting 2% of the total juice volume. The fermentation process was allowed to continue for four days, after which the transformed juice was sterilized using 0.22 μm filtration. Next, the total phenolic content of the samples was measured using the Folin–Ciocalteau method, with gallic acid performing as the standard. The results were expressed in μM Gallic Acid Equivalent (GAE). Notably, the characteristics of blueberry and biotransformed-blueberry juice samples have been described in previous studies [29,30].

The examination and fractionation of the fermented blueberry extract or PEBP using UPLC-MS-QTOF revealed that specific fractions, abundant in bioactive components like gallic acid (GA), Protocatechuic acid (PCA), and catechins (Cat), play a significant role in preserving glucose homeostasis. Commercially obtained Sigma-Aldrich standards (>95% purity) for the major compounds identified in blueberries, namely PCA, GA, and Cat, were utilized in the product extracts [24]. This manuscript will consistently refer to these compounds as the “Oligomeric Mixture of Polyphenols”.

### 2.2. Cell Culture

Human melanoma cell lines HS 294T and A375 were acquired from the American Type Cell Collection (ATCC; Chicago, IL, USA). The cells were maintained in Dulbecco’s Modified Eagle’s Medium (DMEM) (no. 11995065; Gibco, Grand Island, NY, USA) supplemented with Fetal bovine serum (FBS) (10%, *v*/*v*) (no. 30-2020; Gibco, Grand Island, NY, USA) and penicillin/streptomycin (0.05 mg/mL) (J160007; Sigma-Aldrich, Oakville, ON, Canada) at 37 °C in a humidified atmosphere, with 5% CO_2_.

### 2.3. Melanoma Spheroid Cultures

Adherent cells were detached by trypsin and single cells were counted using the countess automated cell counter (Invitrogen, Burlington, ON, Canada). Human melanoma cells were plated on Costar ultra-low attachment plates (no. 07200601; Corning, St. Laurent, QC, Canada) at 10^5^ cells/0.2 mL/well, in the presence or absence of PEBP or OMP in DMEM-F12 (no.12660; Invitrogen, Burlington, ON, Canada), supplemented with 20 μg/mL Epidermal Growth Factor (EGF) (no. E9644; Sigma Aldrich, Oakville, ON, Canada), 20 μg/mL Fibroblast Growth Factor-Basic Human (BFGF) (no. F0291; Sigma Aldrich, Oakville, ON, Canada), 10 mg/mL Insulin solution (no.19278; Sigma Aldrich, Oakville, ON, Canada), 100 mM Sodium pyruvate (S8636; Sigma-Aldrich, Oakville, ON, Canada), 250 mM L-glutamine (no. G6392; Sigma Aldrich, Oakville, ON, Canada), 100 μg/mL hydrocortisone (no. H0135; Sigma Aldrich, Oakville, ON, Canada), and penicillin/streptomycin (1000×) (J160007; Sigma-Aldrich, Oakville, ON, Canada). Cells are grown in these conditions as non-adherent spherical clusters of cells, and spheroids were counted after 2–3 days using light microscopy.

### 2.4. Quantitative Real-Time qPCR

Total RNA was isolated from cell lines following exposure to varying concentrations of PEBP or OMP for 48 h. The extraction was performed using a miRNeasy kit (Qiagen, Toronto, ON, Canada) and quantified using a NanoDrop spectrophotometer (NanoDrop ND1000; Thermo Fisher Scientific, Waltham, MA, USA), in accordance with the manufacturer’s guidelines. The RNA underwent reverse transcription into cDNA using Moloney Murine Leukemia Virus (MMLV) Reverse Transcriptase (Invitrogen, Burlington, ON, Canada, no. 28025013) and miRNA-specific primers obtained from Ambion (Thermo Fisher Scientific, Waltham, MA, USA). Quantification was carried out via real-time PCR using Taqman probes (Applied Biosystems, Burlington, ON, Canada) and FastStart Taq Polymerase (Roche, Mississauga, ON, Canada), following the manufacturer’s protocols. The miRNA PCR reactions underwent incubation at 95 °C for 10 min, followed by 40 cycles of 95 °C for 30 s and 60 °C for 1 min. The internal control utilized was snRNA U6 for normalization in each sample (Applied Biosystems, Burlington, ON, Canada). The cycle threshold values were compared using BIO-RAD CFX96 Manager software #1845000 (Bio-Rad, Mississauga, ON, Canada) to measure the expression of the specified miRNAs, and the relative level was determined using the ΔΔCT method.

### 2.5. Transfection

A total of 2 × 10^5^ cells were allowed to grow to 50% confluence in DMEM medium with FBS. Cells were transfected with either a miR-200c inhibitor, miR-210 mimic or a non-coding control (Ambion, ThermoFisher Scientific), by using lipofectamine 2000 (no.11668019; Life Technologies, Burlington, ON, Canada). After incubation, a passage was completed, and cells were plated in regular-attachment 6-well or ultra-low-attachment 6-well plates. The samples were analysed 48 h post-transfection and measured using RTqPCR.

### 2.6. Western Blots

Protein concentrations were measured using a BCA protein assay kit (no. 23227; Thermo Fisher Scientific, Waltham, MA, USA). Whole-cell lysates were generated through direct lysis with 1× SDS sample buffer, followed by boiling the samples for 10 min at 95 °C. Subsequently, 10 µg of protein samples were loaded into each well in a Mini Gel Tank (Life Technologies, Carlsbad, CA, USA) containing MES Running buffer (20×) (Life Technologies, Carlsbad, CA, USA, no. 1675920). Electrophoresis was conducted at 200 volts for 22 min, and the proteins were then transferred to polyvinylidene fluoride (PVDF) membranes (Invitrogen, Burlington, ON, Canada) using a Trans-Blot Cell (Bio-Rad, Hercules, CA, USA) cooled by a Neslab machine. After membranes were washed with TBST and blocked at room temperature for 1 h in 5% milk in TBST, primary antibodies, including ZEB2, E-cadherin, vimentin, SOCS1, and NF-kB (1:5000 dilution; Abcam, Cambridge, UK), were incubated on the membranes in 5% BSA (Pierce Biotechnology, Thermo Fisher Scientific, no. 23227) in TBST, as per the manufacturer’s recommendations, overnight at 4 °C. After five washes of 15 min each with TBST, the membranes were then exposed to secondary anti-rabbit antibodies (1:10,000 dilution; Ab39368; Jackson ImmunoResearch Laboratories, West Grove, PA, USA) for 1 h at room temperature.

Antibody detection was achieved using ECL solutions (Bio-Rad, Mississauga, ON, Canada), and the membranes were analysed with a VersaDoc (Bio-Rad, Hercules, CA, USA). Normalization was performed using β-actin (Ab8227; Abcam, Cambridge, MA, USA) or Alpha tubulin (Ab4074; Abcam, Cambridge, MA, USA) as a loading control. Densitometry of Western blot results was determined using Image Lab software version 5.0 (Bio-Rad, Hercules, CA, USA).

### 2.7. Patient Samples

Paired tumour/surrounding skin cancer tissues were provided by the Ottawa Hospital Oncology Dept. Ethical approval for using tissue samples was already obtained by Dr. Nessim’s lab. Patients with metastatic melanoma and BRAF(V600E) mutations provided written informed consent for tissue acquisition according to a protocol approved by the Ottawa Hospital Oncology Dept. Patient’s tumours were digested using Miltenyi Biotec’s Tumor Dissociation Kit and gentleMACs Dissociator, following the protocol [31].

### 2.8. Statistical Analysis

All experiments were repeated in triplicate. All values are displayed as mean ± standard error (SEM). Statistical significance was determined by one-way ANOVA, post hoc Tukey test on GraphPad Prism 8 (La Jolla, CA, USA). *p*-values < 0.05 were considered statistically significant.

## 3. Results

### 3.1. Targeting Melanoma Cancer Stem Cells: Inhibition of Sphere Formation by PEBP and OMP

The effect of PEBP and OMP on the formation of melanoma cancer stem cell spheres using the sphere formation assay was investigated. To achieve this, a low-attachment culture condition was utilized, as it favours the formation of CSCs. Different concentrations of PEBP and OMP were added to Hs 294T, A375, and primary melanoma cell lines (MTP), and incubated for 48 h. As shown in Figure 1, the results demonstrated that the spherical structures of the control group had sharp, round edges, and a fusion of spheres was observed. In contrast, a significant reduction in the size of melanoma spheres was observed following treatment with PEBP and OMP, with minimal sphere–sphere fusion. This finding suggests that PEBP and OMP can decrease the size and proportion of cells that can form spheres within the CSC population in melanoma.

### 3.2. Modulatory Effects of PEBP and OMP on the Expression of miR-200c and miR-210 in Melanoma Cancer Stem Cells

To better understand the mechanisms underlying the inhibitory effects of PEBP and OMP on melanoma sphere formation, key miRNAs associated with clinical and pathological aspects of melanoma cancer stem cells were analysed. Total RNA was extracted from A375, HS 294T, and MTP cell lines, and the expression of these miRNAs was assessed using real-time qPCR. We found that after treatment with PEBPs and OMPs, the expression levels of miR-200c were generally higher in melanoma CSCs than in control (**** = *p* < 0.0001) (Figure 2A,D,E). In contrast, the expression of miR-210 was significantly decreased in all melanoma cell lines relative to the control, after treatment with PEBP and OMP (**** = *p* < 0.0001) (Figure 2B,C). These results suggest that the impact of the melanoma CSCs on A375, HS 294T, and on MTP by PEBP and OMP may be partly related to their modulatory activities on miRNAs, as evidenced by the increase in the expression of the tumour-suppressor miR-200c and the decrease in oncogene miRNA-210 expression.

### 3.3. Analyzing the Function of miR-200c and miR-210 in Melanoma Cancer Stem Cells

Furthermore, we investigated the impact of PEBP and OMP on modulating miR-200c overexpression and the downregulation of miR-210 expression in melanoma cancer stem cells. To accomplish this objective, we transfected melanoma cancer stem cells with miR-200c inhibitor and miR-210 mimic and treated them with PEBP and OMP using a transfected reagent.

The expression levels of miR-200c and miR-210 were determined through quantitative real-time PCR. The results demonstrated that PEBP and OMP significantly increased the content of miR-200c when the synthesized miR-200c inhibitor was transfected into the cells. Furthermore, the overexpression of the miR-200c inhibitor resulted in a significant decrease in melanoma cancer stem cells compared to the control, as shown in Figure 3.

On the other hand, treating the transfected cells with PEBP and OMP led to a downregulation of miR-210 mimic in melanoma cancer stem cells, compared to the control. These findings suggest that PEBP and OMP effectively lead to the overexpression of miR-200c inhibitor and the downregulation of miR-210 mimic in melanoma cancer stem cells (Figure 3).

### 3.4. PEBP Inhibits EMT and NF-κB Pathways in Melanoma Cancer Stem Cells

Additionally, we aimed to investigate the mechanisms underlying PEBP’s effect on miR-200c and miR-210 inhibition of EMT and NF-kB in transfected melanoma cancer stem cell lines. A Western blot analysis to identify downstream targets of miR-200c and miR-210 was conducted. The transfection of miR-200c inhibitor resulted in increased ZEB2 and Vimentin and decreased expression of E-cadherin in the melanoma cancer stem cells from the control group. However, when treating the transfected cells with PEBP, our results showed a reversal of this mechanism by increasing the expression of E-cadherin and decreasing the expression of ZEB2 and Vimentin in all melanoma cancer stem cell lines (Figure 4).

Moreover, the transfection of miR-210 by its mimic promoted the activity of NF-κB by negatively regulating SOCS1 in melanoma cancer stem cells. Our findings revealed that PEBP treatment, after down-regulating miR-210 expression, decreased the activation of NF-κB expression and increased the expression of SOCS1, as shown in Figure 5. Collectively, the study suggests that PEBP plays a role in changing the expression of miR-200c and miR-210 by modulating the expression of EMT and NF-κB signalling pathways in melanoma cancer stem cells.

## 4. Discussion

Melanoma is considered one of the deadliest and most aggressive types of skin cancer. In the past decades it has been observed that cases of malignant melanoma have been rapidly increasing [2]. Melanoma can be treatment-resistant to radiotherapy, chemotherapy, immunotherapy and hormonal therapy [2,7]. The surgery option had shown a 95% success rate leading to a complete recovery. However, the surgery is only effective if melanoma is detected in the early stage of development, before metastasis [2,32].

Our group have discovered a bacterium from the blueberry flora (SV-53) that can bio-transform large polyphenolic compounds into small oligomer of polyphenols. The biotransformed mixture designed is Polyphenol-Enriched Blueberry Preparation (PEBP). The process of fermentation and biotranformation greatly enhance the antioxidant potential, when compared to the non-fermented mixture, and endows it with novel anti-inflammatory, anti-diabetic and anticancer properties [20,24,25]. Natural polyphenols were recently found to exhibit various anticancer effects, such as protecting against DNA damage, deregulating crucial cellular signalling pathways, inhibiting cancer stem cell proliferation, and the induction of apoptosis in skin cancer [23,25,33]. The mechanism occurring during fermentation might explain why PEBP showed better inhibitory effects on CSC than the unfermented control [20]. During fermentation, long-chain polyphenols are hydrolysed by microbial enzymes, which render them more bioavailable and biofunctional. Small-chain polyphenols, such as gallic acids, protocatechuic acid and catechins, are more readily absorbed in the digestive tract than long-chain polyphenols [20,24]. Many bioactive compounds are released during fermentation, acting/potentiating in synergy. However, specific active molecules were released through UPLC-QTOF analysis known as Oligomeric Mixture of Polyphenols (OMP), a combination of protocatechuic acid, gallic acid, and catechins, using in the study to shed light on a mechanistic behaviour in support of the hypothesis. Polyphenol preparation contains similar phenolic compounds to those found in green tea and grape seed, known for their chemopreventive and chemotherapeutic properties in many types of cancer [25,34,35,36], hence the strategy to explore the effect of PEBP on skin cancer and CSCs formation.

Sphere formation, or melanospheres in the case of skin cancer melanoma, has been a valuable tool used in stem cell and cancer research to enrich adult stem cells and assess their potential for self-renewal in vitro. The melanoma sphere is a cancer stem cell population with self-renewal and differentiation abilities [37,38,39]. Therefore, characterizing the sphere-forming cells in melanoma carcinoma is crucial to understand the pathogenesis of melanoma carcinoma and developing therapies targeting CSCs [38,40]. Polyphenol compounds such as curcumin, quercetin, and resveratrol have been used to prevent the growth of CSCs, inhibit the formation of mammospheres, and decrease the development of tumours [41]. Accordingly, we have previously shown that PEBP inhibits the proliferation and differentiation of malignant stem cells through modifications to the MAPK/STAT3 signalling pathways, which have significance in controlling CSCs [24,42]. In line with earlier findings, we discovered that PEBP suppressed mammosphere development both ex vivo in breast tumour cell lines and in vitro in the 4T1 and MDA-MB-231 cell lines [42]. Likewise, an earlier work from our group demonstrated that PEBP limits the skin cancer stem cells’ in vitro ability to form spheres [20]. A different study found that polyphenols can significantly reduce mammosphere development in vitro, indicating that they may have negative effects on CSC proliferation and self-renewal in vivo [43]. Our present study has revealed that treating different melanoma cell lines with PEBP and OMP reduced the number and size of sphere formation in melanoma CSCs (Figure 1). Therefore, the PEBP and OMP treatment significantly decreased the self-renewal capacity of melanoma CSCs, preventing proliferation and inhibiting tumour cell regrowth.

Recent study has shed light on how the dysregulation of epigenetic pathways affects gene expression patterns and critical pathways involved in cell proliferation and survival, which can increase the population of CSCs and cancer development [44,45]. Notably, epigenetic alterations, such as miRNA expression, have emerged as fundamental forces in developing and maintaining CSCs [18,46]. Therefore, targeting the epigenetic pathways associated with CSCs could provide new perspectives on cancer therapy. By identifying CSCs and focusing on the underlying epigenetic pathways involved in their development and maintenance, promising strategies for developing effective cancer treatments can be explored [44]. We had previously suggested that PEBP may regulate the stemness of breast cancer, specifically by upregulating miR-145 and downregulating miR-210 expression in vitro [22]. Furthermore, we discovered that PEBP enhances the expression of miR-200b, a miRNA that is normally downregulated in the melanoma cell line [20]. We also reported that the PCA-based combination markedly increased the expression of the tumour suppressor miR-145 in mice tumour samples [24].

Downregulation of miR-200c expression has been shown to promote EMT via the transcriptional factor ZEB2 [46,47,48]. Numerous markers, such as E-cadherin, govern EMT signalling pathways. Expression of E-cadherin is frequently down-regulated in human melanoma [10,46]. The defeat of the epithelial cell-to-cell adhesion by repression of E-cadherin expression is considered one of the hallmarks of activation EMT [9,10,46]. When E-cadherin is absent, downstream signalling pathways of the invasion-metastasis cascade will be activated. These factors regulate E-cadherin expression by binding directly or indirectly to its promoter [10,49]. Studies have shown that miR-200c acts as a tumour suppressor by inhibiting metastasis, cancer cell proliferation, and migration, by targeting various pathways such as EMT [9,46]. miR-200c is a direct target of E-cadherin, and its expression has been linked with metastasis and incision in melanoma cancer tumours [47]. Thus, an increase in the expression of E-cadherin by miR-200c has been reported to suppress melanoma progression [47]. Furthermore, upregulating the level of miR-200c caused an inhibition of ZEB2, which allows E-cadherin to be expressed. Additionally, overexpression of miR-200c is known to down-regulate vimentin, which is known to inhibit cell migration and invasion, as well as to reduce metastatic tumour cells; thus, it leads to reduced EMT [50]. Our previous finding revealed that PEBP impacted the formation of CSCs and decreased the migration and invasion of skin cancer cells by overexpression of miR-200b and inhibition of the transcription factor ZEB1 [20]. In this study, we have demonstrated the functional effects of PEBP and OMP on miR-200c and their related protein genes. Treatment of melanoma cell lines with PEBP and OMP resulted in the overexpression of miR-200c, which caused the negative regulation of ZEB2 and vimentin and up-regulated E-cadherin. These data suggest that miR-200c plays a role in the mechanism by which PEBP or OMP might reverse EMT in melanoma cancer, suppressing tumour progression and metastasis.

On the other hand, elevated expression of miR-210 has been observed in melanoma tissues compared to normal tissue [51,52]. miR-210 expression has been implicated in several cellular processes, including cell cycle regulation, cell survival, differentiation, angiogenesis, proliferation, and apoptosis [51]. Additionally, the increased expression of miR-210 promotes proliferation and reduces apoptosis in melanoma cancer stem cells by targeting the NF-κB pathways which are involved in EMT progression [51,53,54]. Moreover, miR-210 also promotes EMT, and the invasion and migration of melanoma cells, by targeting negative regulators of SOCS1, resulting in constitutive activation of the NF-κB signalling pathway [46,54]. A study has identified that miR-210’s expression promotes the activation of NF-κB signalling via the targeting of SOCS1, further promoting metastasis [54]. Targeting the expression of the NF-κB signalling pathway is crucial to induce and maintain EMT in many cancers, including melanoma, making it a promising target for anti-melanoma cancer therapy [16,54]. We have reported that PEBP modulated the regulation of breast cancer stemness through the downregulation of miR-210 in vitro [22]. Furthermore, polyphenols such as resveratrol induced the expression of SOCS1 in inflammatory diseases [55]. Our results reveal a novel mechanism by which miR-210 sustains constitutive activation of NF-κB signalling. PEBP treatment reduced the expression of miR-210 and decreased expression levels of NF-κB-p65 via negative regulation of the SOCS1 gene, and, further, might reduce EMT invasion and migration in the metastasis of melanoma cells in vitro. The findings also demonstrate how PEBP contributes to the expression of miR-200c and miR-210 in metastatic melanoma cells, and how it can modulate EMT and NF-κB activation through epigenetic modification, thus affecting the EMT pathway.

## 5. Conclusions

The key findings of the current study present a novel insight into the role of polyphenols in general, and small polyphenol oligomers in particular, as a novel complementary supplement against skin cancer and metastasis. Importantly, it strengthens the emerging evidence that miRNAs play a significant role in fine-tuning the host gene expression networks and signalling pathways governing the metastatic properties of melanoma cells. The findings demonstrate that polyphenol compounds such as PEBP can regulate the tumour suppressor miR-200c and oncogene miR-210 involved in the self-renewal of stem cells and EMT pathways.

## Figures and Tables

**Figure 1 biomedicines-12-00193-f001:**
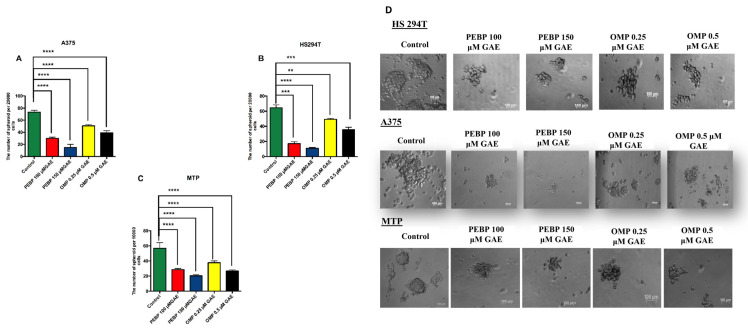
The effect of PEBP and OMP on decreasing the formation of spheres in different melanoma cell lines. The number of sphere formation of (**A**) A375, (**B**) HS 294T, and (**C**) MTP after treatment with either 100 or 150 μM GAE of PEBP or with either 0.25 and 0.5 μM GAE of OMP for 2 days in low-attachment plates and under spheroid culture conditions. (**D**) Representative optical microscopy pictures of 3D-cultured different cell lines. The values A, B, and C represent the mean and standard deviation derived from three separate sets of experiments. All data are presented as mean ± SEM. Significance show as ** = *p* < 0.01 *** = *p* < 0.001, **** = *p* < 0.0001 different from control.

**Figure 2 biomedicines-12-00193-f002:**
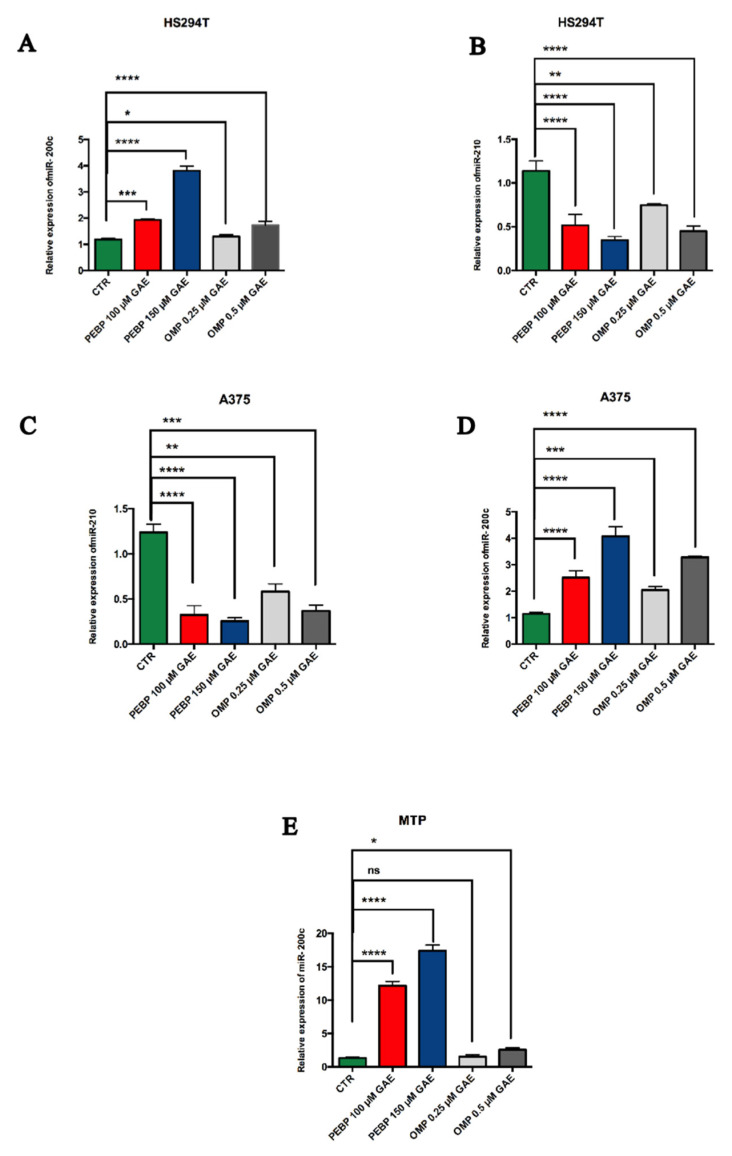
The relative expression of miR-200 and miR-210 in malignant melanoma cancer stem cells. The expression of miR-200c was significantly increased in melanoma cells after treatment with PEBP and OMP (**A**,**D**,**E**), whereas miR-210 expression is decreased in both A375 and HS 294T cells after treatment with PEBP and OMP when compared to untreated control (**B**,**C**). Cells were quantified by qPCR using a TaqMan assay. All values are means of three separate experiments ± SEM. ** Denotes statistical significance at *p* ≤ 0.01 vs. control. Significance is shown as ns: non-significant, * = *p* < 0.05 ** = *p* < 0.01 *** = *p* < 0.001, **** = *p* < 0.0001 different from control for 48 h.

**Figure 3 biomedicines-12-00193-f003:**
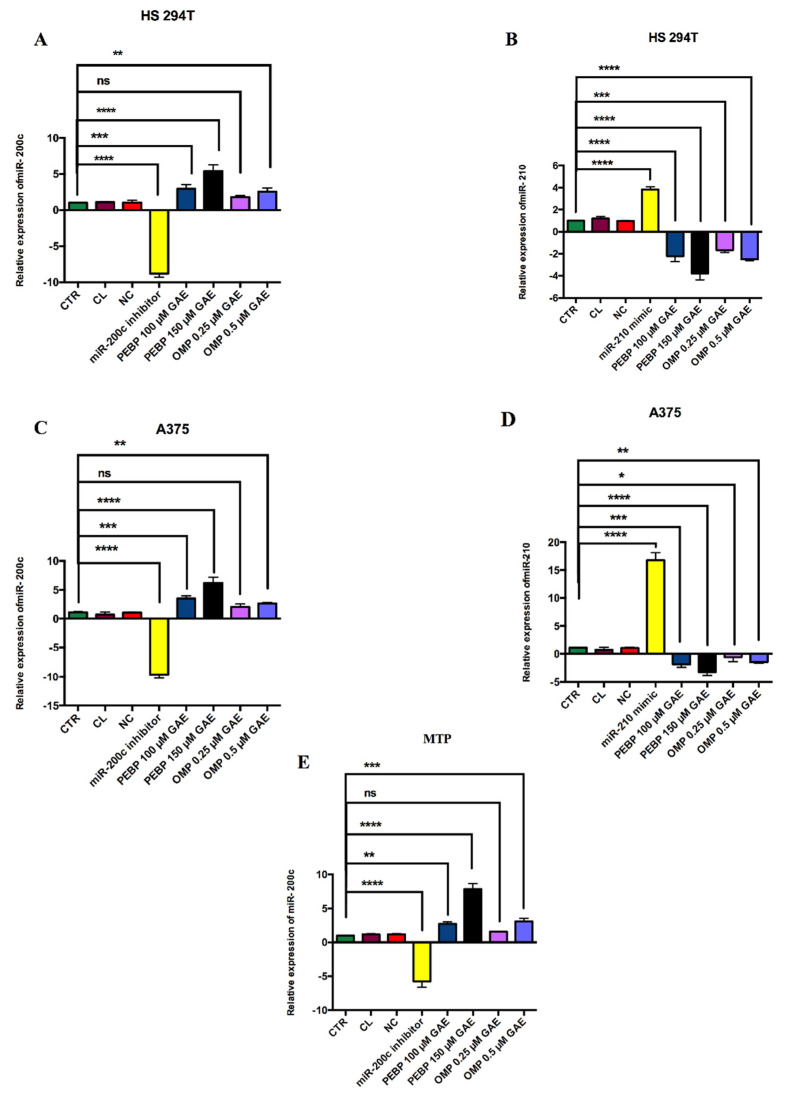
The effect of miR-200c inhibitors and miR-210 mimic in melanoma metastasis cells. Cells were transfected with miR-210 mimic, miR-200c inhibitor, Control Lipofectamine non-target (CL) and noncoding RNA (NC). (**A**,**B**) analysis by qPCR in the transfected HS 294T cells. (**C**,**D**) are A375 transfected cells. (**E**) is MTP transfected cell. Transcript levels were normalized to U6 expression. Error bars represent the mean ± SEM. of three independent experiments. Significance shows as ns: non-significant, * = *p* < 0.05 ** = *p* < 0.01 *** = *p* < 0.001, **** = *p* < 0.0001 different from control.

**Figure 4 biomedicines-12-00193-f004:**
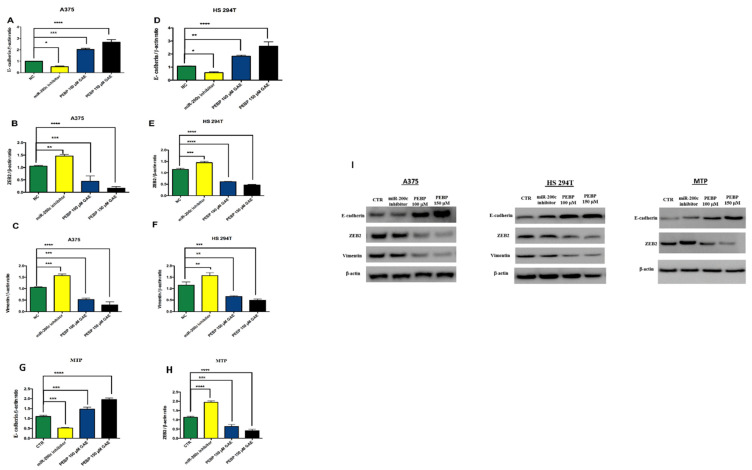
PEBP decreased EMT pathways in A375, HS 294T, and MTP cells in vitro by controlling the expression of E-cadherin, ZEB2 and vimentin. Overexpression of miR-200c increased E-cadherin expression and decreased vimentin and ZEB2 expression in the melanoma cancer stem cell lines. (**A**–**H**) Representative quantifies and normalizes the protein levels using β-actin served as the loading control. (**I**) Representative Western blot images. Error bars represent the mean ± SEM of three independent experiments. * = *p* < 0.05 ** = *p* < 0.01 *** = *p* < 0.001, **** = *p* < 0.0001 different from control for 48 h.

**Figure 5 biomedicines-12-00193-f005:**
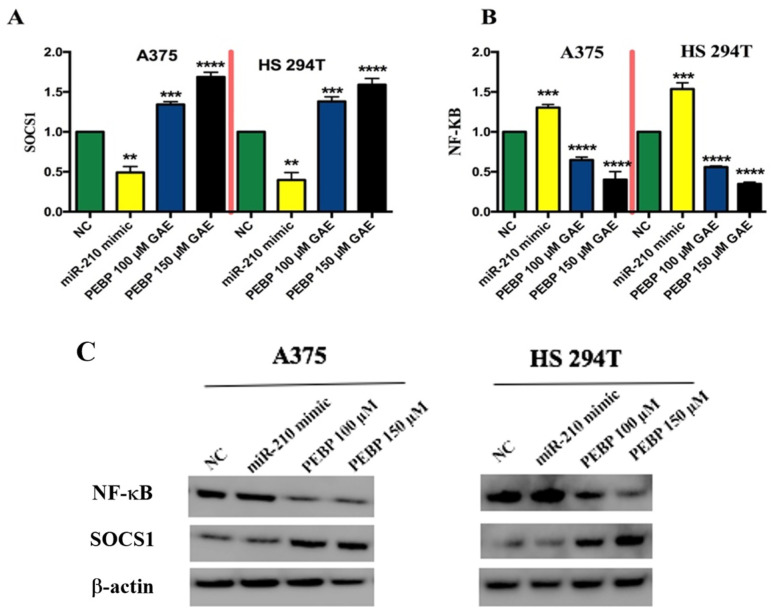
Western blotting of NF-κB/p65 and SOCS1 expression in A375 and HS 294T cells. (**A**,**B**) Representative quantifies and normalizes the protein levels using β-actin. (**C**) Representative Western blot images. Protein expression levels of NF-κB/p65 and SOCS1 were quantified using image lab software version 5.0 and normalized to the corresponding expression levels of β-actin. ** *p* < 0.01, *** *p* < 0.001 and **** = *p* < 0.0001.

## Data Availability

The presented data are available upon request from the corresponding author.

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
