# Peer review of "Role of a Polyphenol-Enriched Blueberry Preparation on Inhibition of Melanoma Cancer Stem Cells and Modulation of MicroRNAs"

_biomedicines, 2024, doi:10.3390/biomedicines12010193_

Round 1
Reviewer 1 Report
Comments and Suggestions for Authors
This paper reports some results concerning the chemopreventative effects of polyphenols derived from fermented blueberries on melanoma termed PEBP obtained with a bacterial strain formerly described by the same group. It is demonstrated that the effect is mediated two miRNAs and subsequent impacting in the Epithelial-to-Mesenchymal Transition (EMT). miR-200c expression is increased and miR-210-3p expression is decreased. Therefore, it is proposed that PEBP might be a promising strategy for skin cancer chemoprevention.
The proposal is interesting, and the research team has experience in the polyphenol-enriched blueberry preparation (PEBP) and the effects in other cancer cells. However, the manuscript contains some unclear points that should be addressed before acceptance. They are as follow:
Major concerns
The enrichment in polyphenols of the blueberries preparation after fermentation should be demonstrated. There is no data concerning this point.
As control, it is used a balanced mixture of polyphenols, including GA, PCA, and catechins to mimic the fermentation process, termed the OMP (Oligomeric Mixture of Polyphenols).
(a) The role of fermentation in the formation of the PEBP from “polymeric” polyphenols should be justified. Monomeric and oligomeric phenols are already found un blueberry extract as in many other plant extracts. The suitable control for the assays would be a non-fermented blueberry extract rather than the OMP.
(b) The ratio among the GA, PCA and catechins concentrations at the OMP should be indicated.
(c) Looking at the figures, the concentration range of PEBP and OMP using GAE units is quite different. This suggests that the PEBP is poorly efficient (around two order of magnitude), In turn, OMP misses anthocyanins, that are considered important antioxidant components of blueberries, and it might justify an eventual greater effect of PEBP. Thus, the use of that PEBP does not improve OMP, and the proposal for its use is greatly weakened.
It is stated that CSC are a subpopulation of cancer cells that significantly contribute to tumor recurrence and differentiation. Material and Methods describes the use of cultures of two well-established melanoma cell lines and a culture of primary melanoma cell. However, results are concerning only CSC without any description about the isolation of CSC from the melanoma cell population. The change from melanoma cells to CSC cells should be described. This is essential, especially at the sections 3.3 and 3.4 .
Minor points
The abbreviation of the nuclear factor kappa-light-chain-enhancer of activated B cells (NFkB) should be corrected throughout the text. Sometimes the Greek kappa letter is omitted, and sometimes it is replaced by the “k” letter.
Line 68-77. It is true that the protective role of polyphenolic compounds in modulating miRNAs in melanoma cancer stem cells is not yet fully understood, but it should be also clear that the antioxidant and anti-inflammatory effects of polyphenols of cancer cells has been largely studied and in addition is mediated by other mechanisms, not only miRNA. Some reference concerning other mechanisms would be added.
Line 113: The Latin name of the bacteria strains (genus) should be written in italics.
Line 131: The HS294T should be named also with capital T.
Lines 142 and 147: Replace u by micro (m Greek letter)
Line 225, Figure 1: “Data is a combination of 3 experiments”. Please, clarify this expression. It is supposed that the A, B and C are the mean and standard deviation of three sets of experiments.
Line 311: Replace b-actin by b-actin.
Lines 330-331: The mechanism occurring during fermentation might explain why PEBP showed better inhibitory effects on CSC than the unfermented control. This is not demonstrated. According to the figures, the effect of PEBP is much lower than OMP, but the unfermented controls are missing. Control without fermentation would be necessary in spite the research team has published several papers and they have experience in the effects of this natural extract on mammalian cells.
Discussion is going further than the results and would be reduced and focused in the data contained in this manuscript.
Conclusion: The expression “critical role” should be ameliorated. It seems exaggerated. It is demonstrated that polyphenols are complementary supplement against skin cancer and metastasis, as indicated, but they are not novel and critical.
Author Response
Dear Reviewer,
Thank you very much for taking the time to review this manuscript. Your constructive comments will help us to improve the quality of our manuscript. Please find the detailed responses below and the corresponding revisions and corrections highlighted in the re-submitted files.
Comments and Suggestions for Authors
This paper reports some results concerning the chemopreventative effects of polyphenols derived from fermented blueberries on melanoma termed PEBP obtained with a bacterial strain formerly described by the same group. It is demonstrated that the effect is mediated two miRNAs and subsequent impacting in the Epithelial-to-Mesenchymal Transition (EMT). miR-200c expression is increased and miR-210-3p expression is decreased. Therefore, it is proposed that PEBP might be a promising strategy for skin cancer chemoprevention.
The proposal is interesting, and the research team has experience in the polyphenol-enriched blueberry preparation (PEBP) and the effects in other cancer cells. However, the manuscript contains some unclear points that should be addressed before acceptance. They are as follow:
Major concerns
The enrichment in polyphenols of the blueberries preparation after fermentation should be demonstrated. There is no data concerning this point.
Response:
We sincerely appreciate your thoughtful comments and suggestions on our paper; your insights are valuable. In addressing this concern, we have already conducted comprehensive characterizations of polyphenol-enriched blueberry preparations in our previous work. Our findings have provided substantial insights into the composition and properties of these preparations.
- DOI: 10.15430/JCP.2021.26.3.162
- DOI: 10.1016/j.jnutbio.2005.05.014
- DOI: 10.1002/jsfa.2142
- DOI: 10.1186/s12967-016-0770-7
As control, it is used a balanced mixture of polyphenols, including GA, PCA, and catechins to mimic the fermentation process, termed the OMP (Oligomeric Mixture of Polyphenols).
- The role of fermentation in the formation of the PEBP from “polymeric” polyphenols should be justified. Monomeric and oligomeric phenols are already found un blueberry extract as in many other plant extracts. The suitable control for the assays would be a non-fermented blueberry extract rather than the OMP.
Response:
This study focuses on investigating the impact of PEBP and OMP on melanoma cancer stem cells. Our prior publication has already demonstrated the significant influence of PEBP on melanoma cancer stem cells compared to non fermented bluberry (NBJ) and control groups. Building on these findings, the current research aims to further elucidate and expand our understanding of the effects of PEBP and OMP on melanoma cancer stem cells.
DOI: 10.1186/s12967-016-0770-7
DOI: 10.15430/JCP.2021.26.3.162
- The ratio among the GA, PCA and catechins concentrations at the OMP should be indicated.
Response:
We appreciate the reviewer's valuable suggestion. In response, we included our recent publication regarding the concentration ratios among gallic acid (GA), protocatechuic acid (PCA), and catechins in the OMP.
DOI:10.3390/ijms24043677
- Looking at the figures, the concentration range of PEBP and OMP using GAE units is quite different. This suggests that the PEBP is poorly efficient (around two order of magnitude), In turn, OMP misses anthocyanins, that are considered important antioxidant components of blueberries, and it might justify an eventual greater effect of PEBP. Thus, the use of that PEBP does not improve OMP, and the proposal for its use is greatly weakened.
Response:
After fermentation, which converts large polyphenols to smaller oligomers, the main polyphenol compounds in OMP are released. OMP, Small oligomers of polyphenols, are known to be better absorbed. It's crucial to recognize that PEBP, enriched with metabolites from the novel probiotic bacterium SV-53, incorporates a broader spectrum of polyphenols. These polyphenols, even in small concentrations, collectively amplify the significance of PEBP. Noteworthy, that the effect of PEBP can also be partly attributed polyphenol-independent mechanisms due to metabolites secreted by the bacterium or the cell wall fragments derived from the bacterium.
It is stated that CSC are a subpopulation of cancer cells that significantly contribute to tumor recurrence and differentiation. Material and Methods describes the use of cultures of two well-established melanoma cell lines and a culture of primary melanoma cell. However, results are concerning only CSC without any description about the isolation of CSC from the melanoma cell population. The change from melanoma cells to CSC cells should be described. This is essential, especially at the sections 3.3 and 3.4 .
Response:
Cancer stem cells (CSCs) were isolated from various cell lines through the spheroid formation assay, serving as a foundational step for subsequent assays in our study. This methodology was chosen for its proven efficacy in enriching and characterizing CSC populations, ensuring the reliability and relevance of our experimental procedures.
Minor points
The abbreviation of the nuclear factor kappa-light-chain-enhancer of activated B cells (NFkB) should be corrected throughout the text. Sometimes the Greek kappa letter is omitted, and sometimes it is replaced by the “k” letter.
Response:
Corrected.
Line 68-77. It is true that the protective role of polyphenolic compounds in modulating miRNAs in melanoma cancer stem cells is not yet fully understood, but it should be also clear that the antioxidant and anti-inflammatory effects of polyphenols of cancer cells has been largely studied and in addition is mediated by other mechanisms, not only miRNA. Some reference concerning other mechanisms would be added.
Response:
We genuinely value the suggestion, and it is true that the antioxidant and anti-inflammatory effects of polyphenols on cancer cells have been extensively investigated. Various epigenetic mechanisms, including DNA methylation and histone modification often mediate these effects. However, it's crucial to emphasize that our specific emphasis in this study is to unravel the impact of polyphenol compounds on melanoma through the lens of miRNA expression. While acknowledging the wider context of polyphenol research, our primary goal is to contribute valuable insights into the intricate relationships between polyphenols and melanoma through the modulation of miRNA expression.
Line 113: The Latin name of the bacteria strains (genus) should be written in italics.
Response:
Corrected. Please see line 114 and line 115
Line 131: The HS294T should be named also with capital T.
Response:
Corrected please see line 131
Lines 142 and 147: Replace u by micro (m Greek letter)
Response:
Corrected. Please see line 142, 143,147.
Line 225, Figure 1: “Data is a combination of 3 experiments”. Please, clarify this expression. It is supposed that the A, B and C are the mean and standard deviation of three sets of experiments.
Response:
Corrected. Please see line 225.
Line 311: Replace b-actin by b-actin.
Response:
Corrected. Please see line 311 and line 291
Lines 330-331: The mechanism occurring during fermentation might explain why PEBP showed better inhibitory effects on CSC than the unfermented control. This is not demonstrated. According to the figures, the effect of PEBP is much lower than OMP, but the unfermented controls are missing. Control without fermentation would be necessary in spite the research team has published several papers and they have experience in the effects of this natural extract on mammalian cells.
Response:
We pervious published the impact of PEBP on CSCs compared to unfermented blueberry.
Reference added. Please see line 330
Discussion is going further than the results and would be reduced and focused in the data contained in this manuscript.
Conclusion: The expression “critical role” should be ameliorated. It seems exaggerated. It is demonstrated that polyphenols are complementary supplement against skin cancer and metastasis, as indicated, but they are not novel and critical.
Reviewer 2 Report
Comments and Suggestions for Authors
29 December 2023
Ms. Ref. No.: biomedicines-2806682
Journal: Biomedicines
Title: Role of a Polyphenol-Enriched Blueberry Preparation on Inhibition of Melanoma Cancer Stem Cells and Modulation of MicroRNAs
Comments:
Thank you for your efforts in writing this article on a very pertinent topic. Moreover, I found the article to be informative and with the potential for further research on this topic in future.
I have some observations where mentioned in the following paragraphs that will be useful for its improvement:
1- In abstract part there is using abbreviation of CSCs in abstract without its complete form. If it is possible its complete form can be mentioned additionally.
2- In the material and method part, fermented blueberries were used. Why fermented forms?
3- According to the result, have they any chemoprevention or chemotherapy or booth of them?
4- What was the main reason for using these cell lines HS 294t and A375? And what about normal cells?
5- Duration time was 48 hours of exposure to different concentrations of PEBP or OMP. What was the criteria for selecting 48 hours of exposure?
6- Moreover, The Following reference can be included in the introduction part for more readability:
· https://doi.org/10.3390/biomedicines11102616
· https://doi.org/10.1007/s12013-023-01171-y
· https://doi.org/10.1007/s00210-023-02551-0
· https://doi.org/10.1007/s11033-021-06928-3
Author Response
Dear Reviewer,
Thank you very much for taking the time to review this manuscript. Your constructive comments will help us to improve the quality of our manuscript. Please find the detailed responses below and the corresponding revisions and corrections highlighted in the re-submitted files.
Comments:
Thank you for your efforts in writing this article on a very pertinent topic. Moreover, I found the article to be informative and with the potential for further research on this topic in future.
I have some observations where mentioned in the following paragraphs that will be useful for its improvement:
- In abstract part there is using abbreviation of CSCs in abstract without its complete form. If it is possible its complete form can be mentioned additionally.
Response:
Corrected. Please see line 14
- In the material and method part, fermented blueberries were used. Why fermented forms?
Response:
The fermentation of blueberries not only releases small polyphenols, enhancing the bioavailability of nutrients for improved absorption, but also holds the potential to amplify the associated health benefits. Through fermentation processes, bioactive compounds undergo transformation, giving rise to metabolites with unique or intensified health-promoting properties. Consequently, elevating blueberries' phenolic content may enhance their anticancer properties and reduce metastatic potential.
In support of this concept, the biotransformation of blueberry juice with a novel strain of bacteria isolated from the blueberry flora has been demonstrated to increase its phenolic content and antioxidant activity. This underscores the significance of fermentation as a strategic approach to unlocking the full health-promoting potential of blueberries.
doi: 10.1186/s12967-016-0770-7
doi: 10.15430/JCP.2021.26.3.162
doi: 10.3390/ijms24043677
https://doi.org/10.1016/j.tifs.2023.01.002
- According to the result, have they any chemoprevention or chemotherapy or booth of them?
Response:
PEBP and OMP both serve as chemopreventive compounds.
- What was the main reason for using these cell lines HS 294t and A375? And what about normal cells?
Response:
We specifically chose two melanoma cancer cell lines with a high propensity for metastasis. The first cell line, A375, exhibits a BRAF mutation, a characteristic observed in around 50% of malignant melanomas. The second cell line, Hs294t, is also highly metastatic but lacks the BRAF mutation. Both cell lines were deliberately selected to serve as an in vitro model, offering a comprehensive perspective for our investigation. Furthermore, these cell lines have been used in our previous studies, providing a basis for comparison or extending existing knowledge.
- Duration time was 48 hours of exposure to different concentrations of PEBP or OMP. What was the criteria for selecting 48 hours of exposure?
Response:
We performed a series of assays, including the lactate dehydrogenase (LDH) assay and MTT assay (tetrazolium reduction assay), to determine the optimal time and dosage of PEBP and OMP administration.
6- Moreover, The Following reference can be included in the introduction part for more readability:
- https://doi.org/10.3390/biomedicines11102616
- https://doi.org/10.1007/s12013-023-01171-y
- https://doi.org/10.1007/s00210-023-02551-0
- https://doi.org/10.1007/s11033-021-06928-3
Response:
Thank you for your suggestion. References have been incorporated into the manuscript.
Round 2
Reviewer 1 Report
Comments and Suggestions for Authors
I appreciate the reply letter. The manuscript has been improved. However, I think that the comparison between the fermented preparation, non-feremented, or just a mixture of polyphenols would be necessary. I can see that the authors have published several papers on this subject, and the focus of this new contribution is on miRNAs.